# A New Insight into the Evaluation of Used Frying Oils Based on the Kinetics of Chemical Changes during the Process

**DOI:** 10.3390/foods12020316

**Published:** 2023-01-09

**Authors:** Seyed Mohammad Reza Moosavi, Reza Farhoosh

**Affiliations:** 1Iran International Standard Organization (INSO), Mashhad P.O. Box 14155-6139, Iran; 2Department of Food Science and Technology, Faculty of Agriculture, Ferdowsi University of Mashhad, Mashhad P.O. Box 91775-1163, Iran

**Keywords:** carbonyls, conjugated dienes, cut-off value, frying, kinetics, polar compounds

## Abstract

Kinetics of change in total polar compounds (TPC), carbonyl value (CV), and conjugated diene value (CDV) were simultaneously investigated during the frying of potato strips in eight oil samples at 170 °C. The CDV at the turning point of the sigmoidal kinetic curves (CDV_T_) with an average of ~19 mmol/L, which was almost equivalent to the TPC and CV of ~14% and ~24 μmol/g, respectively, was found to be as a sensory cut-off value for rejection. To discard frying oils from a toxicological standpoint, the CDV at the mean of the times required to reach the CDV_T_ and the CDV_max_ with an average of ~28 mmol/L (almost equivalent to the TPC and CV of ~22% and ~41 μmol/g, respectively) was determined as the corresponding cut-off value.

## 1. Introduction

Frying is one of the oldest and popular processes broadly used to cook foodstuffs directly in the oils and/or fats of appropriate stability at high temperatures (170–200 °C) and long periods of time. The harsh conditions of the process cause the formation of an extraordinarily complex set of hydrolytic and oxidative products, which may be of a wide variety of molecular weights from volatile to heavy polymeric derivatives. These derivatives not only affect the sensory properties of frying oils and foods being fried but also raise potential toxicological concerns with respect to consumers’ health [1].

Any methods adopted to evaluate the quality of used frying oils should simultaneously consider both sensory and toxicological concerns. The total content of polar compounds (TPC) has commonly been using for decades to toxicologically evaluate used frying oils [2]. It has frequently been shown that TPC increases linearly during the course of frying processes [3,4,5,6] and passes over the cut-off value of 24–27% for rejection [7]. From sensory as well as nutritional points of view, the total carbonyls content known as carbonyl value (CV), including a wide range of secondary oxidation products of crucial importance, has been considered as a valuable measure in the evaluation of used frying oils quality [8]. A sigmoidal pattern has frequently been observed for the change in CV over the typical time ranges of frying processes, i.e., an initial slow increasing phase followed by a rapid rising phase terminated to a maximum level. Afterwards, CV might show constant or reduced quantities [8,9] due essentially to the further degradations of the primary carbonyls to the products of non-carbonyl character and/or higher volatility [8,9,10]. The time required to reach a cut-off CV of 43.5 μmol/g has been reported to be the safety time range of sensory and nutritional interest [8].

Conjugated diene value (CDV), representing a remarkable part of the primary oxidation products, i.e., lipid hydroperoxides, resulting from the double bond shifts in polyunsaturated fatty acids [11], has been shown to correlate well with TPC [4]. On this basis, the time required to reach a cut-off CDV of 29 mmol/L corresponded roughly to a TPC of 24% [8]. Despite the time-consuming and relatively costive preparations involving some chemical reactions in TPC and CV measurements, CDV is simply determined by only reading the UV absorption of a small amount of an oil sample at 232–234 nm [12,13,14]. From a kinetic standpoint, CDV increases initially and then reaches a plateau over frying, due to a balance between the formation rates of conjugated dienes and their dimers resulting from the Diels-Alder reaction [14]. This is in line with the kinetic model of total (conjugated and non-conjugated) lipid hydroperoxides accumulation developed recently by the author [15,16].

Considering the kinetics of the simultaneous formation of polar compounds, carbonyls, and conjugated dienes in a wide range of frying oil samples, the present study aimed to reconsider the evaluation process of frying oils as well as introduce some kinetic parameters to predict their toxicological and also sensory expirations in a simpler, quicker, and less expensive manner.

## 2. Materials and Methods

### 2.1. Materials

A set of eight oil samples, including seven frying oils (A–G) and one non-frying sunflower oil (S), of different compositional properties but passed qualitative and quantitative standards, which were of various vegetable oil sources (individually or as blends), were obtained from seven of big oil factories in Iran. The oil samples were stored at −18 °C until analysis. All of the chemicals and solvents used in the study were of analytical reagent grade and purchased from Merck (Darmstadt, Germany) and Sigma-Aldrich (St. Louis, MO, USA).

### 2.2. Frying Procedure

Peeled and cut (7.0 × 0.5 × 0.3 cm) potatoes were submerged in water until frying. They (20 g) were fried (170 °C) in the eight oil samples (2.0 L) with no replenishment by an electric bench-top fryer (Kenwood DF280, Hampshire, UK). The potato pieces were fried at 15-min intervals for 8 h a day over six days. At given times, ~10 g of the oils were filtered into a screw-cap vial and immediately stored in the dark at 4 °C until analysis. Frying processes were carried out in duplicate.

### 2.3. TPC Measurement

The oil sample (500 mg) was made up to 5 mL with toluene. The solution (1 mL) was pipetted to a 5-mL pipette tip of 15 cm in length packed with 1 g of silica gel 60 (63–100 μm) activated at 160 °C. After soaking in, the pipette tip was eluted (~15 min) by 7 mL (2 × 3.5 mL) of isohexane:diisopropyl ether (85:15 *v*/*v*). TPC (%*w*/*w*) was calculated by Equation (1).
(1)$$\mathrm{TPC}=100\times \left(\frac{w-w_{1}}{w}\right)$$
where *w* and *w*_1_ are the oil weight and the weight of nonpolar components in mg, respectively [17].

### 2.4. CV Determination

Firstly, 2-propanol containing 0.05% (*w*/*w*) of sodium borohydride was refluxed (1 h) and then distilled to remove any trace of carbonyls. 2,4-Dinitrophenylhydrazine (DNPH, 50 mg) was dissolved in 100 mL of 2-propanol containing 3.5 mL of 37% HCl. Oil samples (0.04–1.0 g) were dissolved in 10 mL of 2-propanol containing triphenylphosphine (0.4 mg mL^−1^) to reduce the formation of lipid hydroperoxides. 2,4-Decadienal in 2-propanol (50–500 μM) was used as the standard carbonyl. The standard/oil solutions (1 mL) were mixed with 1 mL of the DNPH solution in a 15-mL test tube. The stoppered test tubes were heated for 20 min at 40 °C. They were cooled in water, and 2% KOH solutions (8 mL) were added. The test tubes were centrifuged at 2000× *g* (Heraeus, Biofuge 13, Germany) for 5 min at room temperature. The absorbance of the upper layers was read at 420 nm by using a spectrophotometer (Jenway 6105 UV-VIS, Essex, UK) against a blank containing all the reagents except that the standard carbonyl solution or the oil was replaced by the solvent alone [18].

### 2.5. CDV Determination

The oil sample was dissolved in hexane (1:600) and its absorbance was read at 234 nm by using a spectrophotometer (Jenway 6105 UV-VIS, Essex, UK) against HPLC grade hexane as blank. An extinction coefficient of 29,000 mol/L was used to calculate millimoles of conjugated dienes per liter [19].

### 2.6. Kinetic Data Analyses

The linear Equation (2) was fitted to the changes in TPC (%*w*/*w*) over the frying time *t* (h).
(2)TPC=a+bt
where *a* and *b* are y-intercept and the rate of change in TPC (*r*_TPC_, /h), respectively.

The method developed recently by the author was employed to calculate the CV-based kinetic parameters [20]. Kinetic curves were drawn by plotting the changes in CV (μmol/g) versus time *t* (h). The sigmoidal Equation (3) was fitted on the kinetic data points of carbonyls accumulation:(3)$$\mathrm{CV}=a+\frac{b}{1+e^{\left(\frac{c-t}{d}\right)}}$$
where *a*, *b*, *c*, and *d* are the equation parameters. The finite value CV_max_, where the rate of carbonyls accumulation reaches zero at infinity, equals *a* + *b*. At the equation’s turning point with the coordinates *t*_T_ = *c* and CV_T_ = *a* + 0.5*b*, the rate of carbonyls accumulation (μmol/g h) reaches the maximum value *r*_max_ = 0.25*b*/*d*. Its normalized form (*r*_max_/CV_max_ = *r*_n_, /h) is calculated from the ratio *b*/4*d*(*a* + *b*). The time at which carbonyls practically approach CV_max_ (*t*_max_, h) is obtained from *c* + 2*d*. The value of *t*_43.5_ (h) is given by Equation (4):(4)$$t_{43.5}=c-d\ln\left(\frac{a+b-43.5}{43.5-a}\right)$$

Kinetic curves of the accumulation of conjugated dienes were drawn by plotting the changes in CDV (mmol/L) versus time *t* (h). The sigmoidal Equation (5) was fitted on the kinetic data points [15]:(5)$$\mathrm{CDV}=\frac{a}{b+e^{a\left(c-t\right)}}$$
where *a*, *b*, and *c* are the equation parameters. The finite value CDV_max_ is calculated from the ratio *a*/*b*. At the equation’s turning point with the coordinates *t*_T_ = (*ac −* ln*b*)/*a* and CDV_T_ = 0.5*a*/*b*, the rate of the accumulation of conjugated dienes (mmol/L h) reaches the maximum value *r*_max_ = 0.25*a*^2^/*b*. Its normalized form (*r*_max_/CDV_max_ = *r*_n_, /h) is given by 0.25*a*. The time at which the content of conjugated dienes practically approaches CDV_max_ (*t*_max_, h) is obtained from the ratio (2 + *ac −* ln*b*)/*a*.

### 2.7. Statistical Analysis

All determinations were carried out in triplicate and data were subjected to analysis of variance (ANOVA). ANOVA and regression analyses were performed according to the MStatC and SlideWrite software version 7.0. Significant differences between means were determined by Duncan’s multiple range tests. *p* values less than 0.05 were considered statistically significant.

## 3. Results and Discussion

### 3.1. Kinetics of Change in Polar Compounds

The unfried oil samples contained TPCs of 3.4–11.0% (Table 1), indicating a wide variation in their initial quality. Fresh and refined vegetable oils have been shown to contain TPCs of 0.4–6.4% [21], although higher contents up to 14% have also been reported [22].

As expected, TPCs increased linearly during the frying process with high coefficients of determination (R^2^ > 0.97) (Figure 1A). As shown in Table 1, the values of *r*_TPC_ as a quantitative measure of the innate potency of an oil in preventing the formation of polar compounds was less than 1 /h for the frying oils A–G (1.63/h for the sunflower oil S). Accordingly, the frying oil B was of the highest toxicological quality, followed by the oils D, F, E, C~A~G, and S. However, a different order of frying performance was obtained when considering the time required to reach the cut-off TPCs of 24% (*t*_24_) or 27% (*t*_27_): B > D > E~A~F > G > C > S. This is due naturally to the different initial quality of the oils with respect to the TPC_0_ values shown in Table 1. In other words, in addition to the innate potency (the *r*_TPC_ value), the initial number of polar compounds is of crucial importance in evaluating the overall toxicological quality of a frying oil.

### 3.2. Kinetics of Change in Carbonyls

Various CVs ranged from 7.8 to 22.8 μmol/g were found in the unfried oil samples (Table 2). Well-refined oils have been reported to have 0.5–2.0 μmol/g of carbonyls [23]. The sigmoidal Equation (3) appropriately fitted (R^2^ > 0.98) the changes in CVs over the frying time at 170 °C (Figure 1B). Table 2 presents the kinetic data characterizing the time change pattern of CV during the frying process of the oil samples.

After attaining the *t*_max_ values of 23.4–54.0 h, CVs reached their maximum values of 44–352 μmol/g (Table 2). The level of CV_max_ is governed by a balance between the rate of carbonyls formation, which can be represented by the value of *r*_max_ on average, and the rate of their degradation. These rates are in turn affected by the chemical composition of oxidizing systems as well as some extrinsic factors such as temperature [20]. Therefore, the *r*_n_ values, ranged from 0.0127 to 0.0528 /h, can be employed to show the innate susceptibility of frying oils to their CV variations. On this basis, the oils of higher sensory quality were in the order of D > E > G > A > C ~ B ~ F > S, which was not in the same performance order as the *r*_TPC_ value. This order was also not highly consistent with those from the time parameters *t*_T_, *t*_max_, or *t*_43.5_. As mentioned above, this can be attributed to the different initial qualities of the oils with respect to the CV_0_ values (Table 2).

As for the cut-off CV of 43.5 μmol/g, it is noteworthy to mention that it was lower than all the CV_max_ levels but higher than the quantities of CV_T_ in the oils B, C, D, E, and G. Furthermore, the calculated CVs at the TPCs of 24 and 27% exceeded that level in some cases (Table 2), indicating their undesirable sensory but acceptable toxicological quality. A more detailed analysis of the matter is provided in the following section.

### 3.3. Kinetics of Change in Conjugated Dienes

The oils were of quite different initial quality with respect to the values of CDV_0_ ranged from 2.4 to 17.1 mmol/L (Table 3). The sigmoidal Equation (5) fitted well (R^2^ > 0.97) the changes in the contents of conjugated dienes over the frying time at 170 °C (Figure 1C). Table 3 presents the kinetic data characterizing the time change pattern of CDV during the frying process of the oil samples.

Distinct from the plateau-reaching pattern of the CDV during frying [14], total lipid hydroperoxides have been shown to significantly decompose after a steep rising phase and reaching their maximum concentration [16]. The decomposition phase is accompanied by the generation of a wide range of secondary oxidation products of dramatic negative impacts on sensory attributes and potential toxicity of the system [24]. The comparison between the contents of conjugated dienes and total lipid hydroperoxides in monitoring the quality of used frying oils in previous research demonstrated that the onset of the total lipid hydroperoxides decomposition phase corresponded to an average CDV of ~18.4 mmol/L [4]. Interestingly, as shown in Table 3, this was in the range of the CDV_T_ values for the frying oils studied (14.5–22.7 mmol/L) with an average of 19.2 mmol/L, which is about two-thirds of the cut-off CDV of 29 mmol/L corresponded to the TPC of 24% [8]. Thus, the time parameter *t*_T_ can be adopted as a measure of frying stability of the oils: S~D > A~F > B~G ≥ C > E. Such an order is naturally different from that according to the unifying parameter *r*_n_ (F > E > C > D > B~G > S~A), which solely addresses the innate potency of the oil samples regardless of their initial quality with respect to the CDV_0_ values (Table 3).

The CV and TPC calculated at the *t*_T_ values ranged from 14.0 to 34.8 μmol/g and from 9.1 to 16.2% for the frying oils with the averages of 23.6 μmol/g and 13.5%, respectively (Table 3). These are much lower than the cut-off CV and TPC values of 43.5 μmol/g and 24–27%, respectively, to discard frying oils. Considering the values of CDV_max_ (Table 3), the cut-off CDV of 29 mmol/L is roughly out of the range with an average of 38.4 mmol/L for the frying oils. However, it falls in the range of the CDV calculated at (*t*_T_ + *t*_max_)/2 with an average of 28.1 mmol/L for the frying oils. Moreover, the cut-off CV and TPC values of 43.5 μmol/g and 24–27% fall in the ranges of the CV and TPC values calculated at (*t*_T_ + *t*_max_)/2 with the averages of 41.2 μmol/g and 21.6%, respectively. In accordance, the CDV at *t*_T_ (CDV_T_) and at (*t*_T_ + *t*_max_)/2 from the sigmoidal Equation (5) can be employed as the sensory and toxicological measures, respectively, to reject frying oils. Plus, the two time parameters *t*_T_ and (*t*_T_ + *t*_max_)/2 can practically be used as the corresponding stability times.

## 4. Conclusions

The present study revealed that the cut-off TPC, CV, or CDV values of 24–27% (~22% in this study), 43.5 μmol/g (~41 μmol/g in this study), or 29 mmol/L (~28 mmol/L in this study), respectively, are almost equivalent measures to reject the used frying oils from a toxicological standpoint. For the sensory rejection of used frying oils, much lower average values of such quantities (about 14%, 24 μmol/g, or 19 mmol/L, respectively) should be considered. Most importantly, the simpler, quicker, and less expensive CDV method provided the CDV at *t*_T_ (CDV_T_) and (*t*_T_ + *t*_max_)/2 of predicting value regarding the sensory and toxicological expirations, respectively, for any individual frying oil. Moreover, some valuable kinetic parameters, especially *r*_n_, were adopted to comparatively evaluate the innate potency of frying oils against oxidative deteriorations.

## Figures and Tables

**Figure 1 foods-12-00316-f001:**
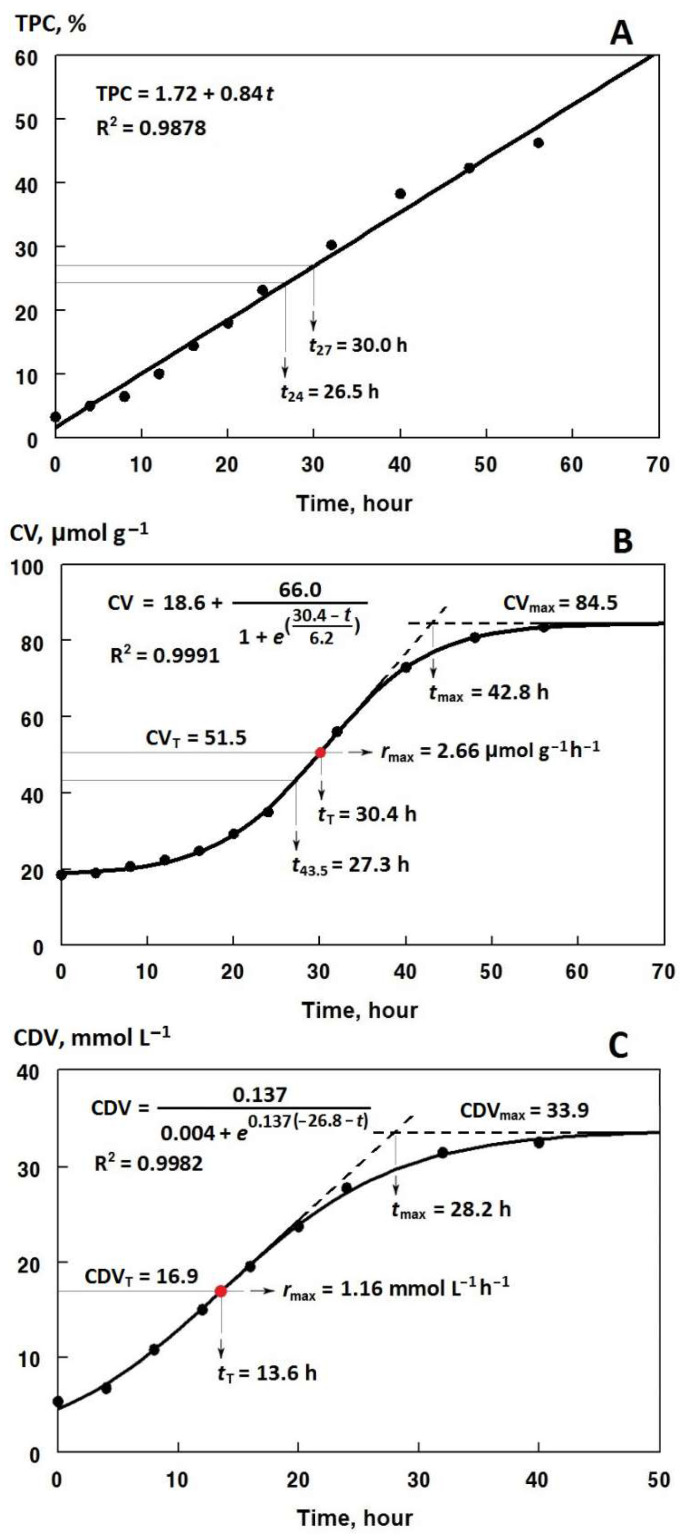
Kinetic curves of the accumulation of polar compounds (TPC, **A**), carbonyls (CV, **B**), and conjugated dienes (CDV, **C**) during the frying of the oil sample A at 170 °C, and the kinetic parameters from the linear Equation (2) and the sigmoidal Equations (3) and (5) fitted on the whole range of the data points. *t*_24_ and *t*_27_: the time required to reach TPCs of 24 or 27%, respectively; CV_T_/CDV_T_: CV/CDV at the turning point of the sigmoidal equations with the x-coordinates *t*_T_; CV_max_/CDV_max_: the maximum CV/CDV levels attained at the values of *t*_max_; *r*_max_: maximum rate of CV/CDV changes; *t*_43.5_: the time required to reach a CV of 43.5 μmol/g.

**Table 1 foods-12-00316-t001:** The parameters resulting from the linear changes in the total content of polar compounds (TPC, %) over the frying time (*t*, h) at 170 °C for the oil samples A–S.

	The Oil Samples							
Parameter	A	B	C	D	E	F	G	S
TPC_0_ ^1^	3.35 ± 0.15 ^g^	5.77 ± 0.22 ^d^	11.0 ± 0.23 ^a^	7.47 ± 0.20 ^c^	4.16 ± 0.14 ^f^	8.54 ± 0.13 ^b^	7.76 ± 0.36 ^c^	5.04 ± 0.19 ^e^
*r* _TPC_ ^2^	0.84 ± 0.01 ^b^	0.40 ± 0.01 ^f^	0.83 ± 0.03 ^b^	0.50 ± 0.00 ^e^	0.77 ± 0.01 ^c^	0.62 ± 0.01 ^d^	0.85 ± 0.00 ^b^	1.63 ± 0.07 ^a^
*t* _24_ ^3^	26.5 ± 0.9 ^c^	49.5 ± 0.4 ^a^	20.3 ± 0.5 ^e^	34.0 ± 0.7 ^b^	27.1 ± 0.7 ^c^	26.4 ± 0.6 ^c^	23.0 ± 0.2 ^d^	11.1 ± 0.2 ^f^
*t* _27_ ^3^	30.0 ± 0.9 ^c^	57.1 ± 0.5 ^a^	23.9 ± 0.7 ^e^	40.0 ± 0.7 ^b^	31.0 ± 0.7 ^c^	31.2 ± 0.7 ^c^	26.6 ± 0.2 ^d^	12.9 ± 0.1 ^f^

Means ± SD (standard deviation) within a row with the same lowercase letters are not significantly different at *p* < 0.05. ^1^ TPC (%) at *t* = 0, representing the initial quality of the oils; ^2^ The rate of change in TPC (/h), representing the innate potency of the oils in preventing the formation of polar compounds; ^3^ The time (h) required to reach a TPC of 24 or 27%.

**Table 2 foods-12-00316-t002:** The parameters resulting from the sigmoidal changes (Equation (3)) of carbonyl value (CV) over the frying time (*t*, h) at 170 °C for the oil samples A–S.

	The Oil Samples							
Parameter	A	B	C	D	E	F	G	S
CV_0_ ^1^	18.5 ± 1.0 ^b^	12.6 ± 0.3 ^c^	9.8 ± 0.1 ^d^	17.3 ± 0.1 ^b^	10.0 ± 0.2 ^d^	9.2 ± 0.0 ^e^	7.8 ± 0.0 ^f^	22.8 ± 0.4 ^a^
CV_T_ ^2^	51.5 ± 0.8 ^c^	40.0 ± 1.6 ^d^	35.8 ± 1.2 ^e^	28.7 ± 0.0 ^f^	22.4 ± 2.9 ^g^	70.8 ± 0.5 ^b^	19.7 ± 2.9 ^g^	194 ± 7 ^a^
CV_max_ ^3^	84.5 ± 1.4 ^c^	65.0 ± 2.8 ^d^	66.3 ± 3.7 ^d^	44.0 ± 0.2 ^f^	55.2 ± 0.3 ^e^	129 ± 1 ^b^	61.7 ± 0.5 ^d^	352 ± 14 ^a^
CV at								
TPC ^4^ = 24%	41.6 ± 1.0 ^c^	65.0 ± 2.8 ^a^	48.1 ± 2.4 ^b^	32.8 ± 0.1 ^e^	37.2 ± 0.1 ^d^	25.4 ± 2.9 ^f^	44.1 ± 0.4 ^b^	39.3 ± 0.9 ^c^
TPC ^4^ = 27%	50.5 ± 0.6 ^c^	65.0 ± 2.8 ^a^	54.7 ± 1.9 ^b^	35.7 ± 0.1 ^f^	40.3 ± 0.1 ^e^	40.3 ± 2.3 ^e^	47.6 ± 0.5 ^d^	41.3 ± 1.2 ^e^
*t* _T_ ^5^	30.4 ± 0.5 ^b^	14.1 ± 0.4 ^d^	15.3 ± 1.2 ^d^	26.5 ± 0.1 ^c^	12.1 ± 3.1 ^de^	37.1 ± 0.6 ^a^	6.9 ± 1.9 ^e^	29.8 ± 1.2 ^b^
*t* _max_ ^6^	42.8 ± 0.1 ^c^	23.4 ± 0.6 ^f^	27.1 ± 3.6 ^ef^	54.0 ± 0.5 ^a^	43.0 ± 0.5 ^c^	47.3 ± 0.3 ^b^	31.2 ± 0.6 ^e^	38.3 ± 1.3 ^d^
*t* _43.5_ ^7^	27.3 ± 0.4 ^d^	15.5 ± 0.2 ^g^	18.3 ± 1.0 ^f^	83.5 ± 3.7 ^a^	35.7 ± 0.3 ^b^	31.9 ± 1.0 ^c^	22.5 ± 0.3 ^e^	14.3 ± 0.9 ^g^
*r* _max_ ^8^	2.66 ± 0.16 ^c^	2.70 ± 0.10 ^c^	2.64 ± 0.33 ^c^	0.56 ± 0.01 ^f^	1.06 ± 0.02 ^e^	5.74 ± 0.53 ^b^	1.73 ± 0.01 ^d^	18.6 ± 0.6 ^a^
*r* _n_ ^9^	0.0315 ± 0.0014 ^c^	0.0416 ± 0.0005 ^b^	0.0401 ± 0.0072 ^b^	0.0127 ± 0.0001 ^f^	0.0192 ± 0.0002 ^e^	0.0445 ± 0.0040 ^b^	0.0281 ± 0.0003 ^d^	0.0528 ± 0.0011 ^a^

Means ± SD (standard deviation) within a row with the same lowercase letters are not significantly different at *p* < 0.05. ^1^ Carbonyl value (μmol/g) at *t* = 0, representing the initial quality of the oils; ^2^ Carbonyl value (μmol/g) at the turning point of the sigmoidal equation; ^3^ The maximum level of CV (μmol/g) attained during the frying process; ^4^ The total content of polar compounds (%); ^5^ The time (h) required to reach CV_T_; ^6^ The time (h) at which carbonyls practically approach CV_max_; ^7^ The time (h) required to reach a CV of 43.5 μmol/g; ^8^ Maximum rate of carbonyls accumulation (μmol/g h); ^9^ Normalized *r*_max_ (/h).

**Table 3 foods-12-00316-t003:** The parameters resulting from the sigmoidal changes (Equation (5)) of conjugated diene value (CDV, mmol/L) over the frying time (*t*, h) at 170 °C for the oil samples A–S.

	The Oil Samples							
Parameter	A	B	C	D	E	F	G	S
CDV_0_ ^1^	5.4 ± 0.2 ^fg^	6.2 ± 0.5 ^eg^	13.1 ± 0.2 ^b^	6.5 ± 0.1 ^e^	17.1 ± 0.3 ^a^	11.6 ± 0.0 ^c^	9.2 ± 0.1 ^d^	2.4 ± 0.1 ^h^
CDV_T_ ^2^	16.9 ± 0.2 ^d^	14.5 ± 0.1 ^e^	20.7 ± 0.1 ^c^	21.1 ± 0.0 ^b^	22.7 ± 0.1 ^a^	16.9 ± 0.1 ^d^	21.5 ± 1.1 ^abc^	17.9 ± 1.1 ^d^
CDV_max_ ^3^	33.9 ± 0.3 ^d^	28.9 ± 0.1 ^e^	41.4 ± 0.2 ^c^	42.3 ± 0.1 ^b^	45.3 ± 0.2 ^a^	33.8 ± 0.3 ^d^	43.1 ± 2.2 ^abc^	35.8 ± 2.1 ^d^
*t* _T_ ^4^	13.6 ± 0.2 ^b^	12.2 ± 0.2 ^c^	10.1 ± 0.2 ^d^	17.4 ± 0.1 ^a^	9.5 ± 0.0 ^e^	13.8 ± 0.7 ^b^	11.5 ± 0.8 ^cd^	17.8 ± 0.3 ^a^
*t* _max_ ^5^	28.2 ± 0.4 ^f^	29.2 ± 0.7 ^ef^	35.5 ± 0.6 ^d^	38.0 ± 0.0 ^c^	42.5 ± 0.8 ^b^	50.2 ± 2.4 ^a^	29.4 ± 2.0 ^ef^	32.0 ± 1.6 ^e^
*r* _max_ ^6^	1.16 ± 0.01 ^b^	0.86 ± 0.03 ^d^	0.82 ± 0.01 ^d^	1.03 ± 0.00 ^c^	0.69 ± 0.01 ^e^	0.47 ± 0.02 ^f^	1.20 ± 0.02 ^ab^	1.26 ± 0.04 ^a^
*r*_n_ ^7^	0.0342 ± 0.0005 ^a^	0.0296 ± 0.0009 ^b^	0.0197 ± 0.0003 ^d^	0.0243 ± 0.0001 ^c^	0.0152 ± 0.0004 ^e^	0.0137 ± 0.0006 ^f^	0.0280 ± 0.0009 ^b^	0.0353 ± 0.0012 ^a^
At *t*_T_								
CV ^8^	22.7 ± 0.4 ^e^	34.8 ± 0.3 ^b^	23.0 ± 1.4 ^de^	23.8 ± 0.0 ^d^	19.7 ± 0.2 ^f^	14.0 ± 0.1 ^g^	27.5 ± 0.3 ^c^	53.5 ± 3.2 ^a^
TPC ^9^	13.2 ± 0.6 ^c^	9.1 ± 0.1 ^e^	15.6 ± 0.1 ^b^	15.7 ± 0.4 ^b^	10.6 ± 0.6 ^d^	16.2 ± 0.3 ^b^	14.2 ± 0.2 ^c^	35.0 ± 0.0 ^a^
At (*t*_T_ + *t*_max_)/2								
CDV	24.8 ± 0.2 ^c^	21.2 ± 0.1 ^d^	30.3 ± 0.1 ^b^	30.9 ± 0.1 ^a^	33.1 ± 0.2 ^a^	24.7 ± 0.2 ^c^	31.5 ± 1.6 ^ab^	26.2 ± 1.5 ^c^
CV ^8^	30.3 ± 1.0 ^e^	55.2 ± 1.4 ^b^	52.9 ± 2.1 ^b^	29.3 ± 0.0 ^e^	36.2 ± 0.1 ^d^	43.8 ± 2.2 ^c^	40.7 ± 0.2 ^c^	112 ± 11 ^a^
TPC ^9^	19.3 ± 0.7 ^f^	12.5 ± 0.0 ^g^	26.1 ± 0.5 ^b^	20.8 ± 0.4 ^e^	23.2 ± 0.5 ^c^	27.5 ± 0.4 ^b^	21.8 ± 0.2 ^d^	46.5 ± 0.6 ^a^

Means ± SD (standard deviation) within a row with the same lowercase letters are not significantly different at *p* < 0.05. ^1^ Conjugated diene value (mmol/L) at *t* = 0, representing the initial quality of the oils; ^2^ Conjugated diene value (mmol/L) at the turning point of the sigmoidal equation; ^3^ The maximum CDV (mmol/L) attained during the frying process; ^4^ The time (h) required to reach CDV_T_; ^5^ The time (h) at which conjugated dienes practically approach CDV_max_; ^6^ Maximum rate of the accumulation of conjugated dienes (mmol/L h); ^7^ Normalized *r*_max_ (/h); ^8^ Carbonyl value (μmol/g); ^9^ The total content of polar compounds (%).

## Data Availability

All the necessary data generated and/or analysed during the current study are included in this published article and its additional information, if needed, are available from the corresponding author on reasonable request.

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
