# Peer review of "A New Insight into the Evaluation of Used Frying Oils Based on the Kinetics of Chemical Changes during the Process"

_foods, 2023, doi:10.3390/foods12020316_

Round 1

Reviewer 1 Report

Line 39-40: There is something technically wrong with the sentence.

Line 62-67 (in 2.1 Materials): There is not enough information about seven frying oils (A – G) and one non-frying sunflower oil sample (S). "Different commercial brands were purchased from a local market" - it is not a scientific approach. This is not a completely justified way of securing samples for scientific research.

Line 81, 104, 110, 118, 122: There is something technically wrong with the way the equations are displayed.

Line 75-134: In connection with the first part of the previous comment, the authors did not plan to examine the composition of samples of seven frying oils (A – G) and one non-frying sunflower oil sample (S) in the experimental design. In connection with the first part of the previous comment, the authors did not plan to examine the composition of samples of seven frying oils (A – G) and one non-frying sunflower oil sample (S) in the experimental design. For example, the composition of fatty acids and the content and composition of minor compounds of these oils, whether they contain/do not contain antioxidants, etc., are not known.

Because of the previous comment, the results and discussion and conclusions are incomplete and poor.

The authors are supported for the choice of the topic of this manuscript and the experiments performed.

The obtained results have a very large practical application.

But the characteristics, composition, etc., of the oils used for frying must be known in more detail.

Author Response

Comment: Line 39-40: There is something technically wrong with the sentence.

Response: The problem was eliminated in the revised manuscript.

Comment: Line 62-67 (in 2.1 Materials): There is not enough information about seven frying oils (A – G) and one non-frying sunflower oil sample (S). "Different commercial brands were purchased from a local market" - it is not a scientific approach. This is not a completely justified way of securing samples for scientific research.

Response: The sentence was improved to give information in a more scientific manner.

Comment: Line 81, 104, 110, 118, 122: There is something technically wrong with the way the equations are displayed.

Response:The equations have no problem in the original manuscript and there is probably something wrong with the software when turning the manuscript into the journal format.

Comment: Line 75-134: In connection with the first part of the previous comment, the authors did not plan to examine the composition of samples of seven frying oils (A – G) and one non-frying sunflower oil sample (S) in the experimental design. For example, the composition of fatty acids and the content and composition of minor compounds of these oils, whether they contain/do not contain antioxidants, etc., are not known.

Response: With respect to the reviewer’s comment, kindly in the author’s opinion, who has been working and reviewing more than twenty years regarding lipid chemistry and related topics, it should be mentioned that having some extra information about the chemical composition of the oil samples could undoubtedly be valuable but the lack of them does not hurt at all the scientific quality of the manuscript of this kind. The present paper had not aimed to compare the oxidative stability of the oil samples based on their chemical compositions. In fact, it was just enough to ensure that we had a set of oil samples of different compositional properties resulting from various vegetable oil sources (individually or as blends), which accordingly we supplied them from seven of different oil factories across the country. Moreover, quite wide variety in the quality indicators as well as in the kinetic data patterns shown in the manuscript clearly affirm a broad diversity in the chemical composition of the oils examined. Hence, considering the kinetics of the simultaneous formation of polar compounds, carbonyls, and conjugated diene hydroperoxides in such a wide range of oil samples, the present study aimed to reconsider the evaluation process of frying oils as well as introduce some kinetic parameters to predict their toxicological and also sensory expirations in a simpler, quicker, and less expensive manner, the issue that fortunately this study has been able to successfully make it practical.

Reviewer 2 Report

The topic of the article covers importnat point connected with safety aspects of frying oils during frying process. The proposition to measure other compounds instead of TPC is oryginal, however nowadays are quick tests to measure this parameter so it is not a problem any longer. The experiment was well planned so obtaind results can be used as predictors of frying oil stability.

It would be nice to involve more information about used frying oils - especially that there were significant differences between them. So please add additional table with charactersitics of these oils. Please add as well more detailed information about CV and CDV determination (type of equipment used ect.) There are no data about sensory analyses, while in the text and conclusions are information about sensory expirations. Please add information about sensory analyses and results.

Author Response

Comment: It would be nice to involve more information about used frying oils - especially that there were significant differences between them. So please add additional table with charactersitics of these oils.

Response: With respect to the reviewer’s comment, kindly in the author’s opinion, who has been working and reviewing more than twenty years regarding lipid chemistry and related topics, it should be mentioned that having some extra information about the chemical composition of the oil samples could undoubtedly be valuable but the lack of them does not hurt at all the scientific quality of the manuscript of this kind. The present paper had not aimed to compare the oxidative stability of the oil samples based on their chemical compositions. In fact, it was just enough to ensure that we had a set of oil samples of different compositional properties resulting from various vegetable oil sources (individually or as blends), which accordingly we supplied them from seven of different oil factories across the country. Moreover, quite wide variety in the quality indicators as well as in the kinetic data patterns shown in the manuscript clearly affirm a broad diversity in the chemical composition of the oils examined. Hence, considering the kinetics of the simultaneous formation of polar compounds, carbonyls, and conjugated diene hydroperoxides in such a wide range of oil samples, the present study aimed to reconsider the evaluation process of frying oils as well as introduce some kinetic parameters to predict their toxicological and also sensory expirations in a simpler, quicker, and less expensive manner, the issue that fortunately this study has been able to successfully make it practical.

Comment: Please add as well more detailed information about CV and CDV determination (type of equipment used ect.).

Response: It was done.

Comment: There are no data about sensory analyses, while in the text and conclusions are information about sensory expirations. Please add information about sensory analyses and results.

Response: As explained in the text, the indicator carbonyl value, which shows the amount of carbonyls as one of the most important compounds to judge about the sensory quality of used frying oils, was adopted to represent the sensory analysis of the oils.  

Round 2

Reviewer 1 Report

Considernig the cooment:

Comment: Line 75-134: In connection with the first part of the previous comment, the authors did not plan to examine the composition of samples of seven frying oils (A – G) and one non-frying sunflower oil sample (S) in the experimental design. For example, the composition of fatty acids and the content and composition of minor compounds of these oils, whether they contain/do not contain antioxidants, etc., are not known.

Without doubting the competence and experience of the author, thank you for the explanation.

A potential reader of the article may not be knowledgeable enough to understand these reasons without further explanation.

Based on the explanations they gave, the authors should supplement the manuscript, and briefly give such or similar explanations, e.g. in the introduction and/or in Material & Methods.

Author Response

Many thanks for the constructive comment of the respected reviewer. A supplementary explanation on the sample oils had been added to the section Materials and Methods in the revised manuscript that I think the respected reviewer has not seen that probably. Please take a look at the revised manuscript submitted before. It should be convincing. 

Kind regards